# Active SMA Fibers’ Effect on the Pullout Behavior of a Steel Bar Embedded in Concrete

**DOI:** 10.3390/ma16051947

**Published:** 2023-02-27

**Authors:** Eunsoo Choi, Bui Thanh Nhan, Jongkwon Choi

**Affiliations:** Department of Civil and Environmental Engineering, Hongik University, Seoul 04066, Republic of Korea

**Keywords:** SMA crimped fiber, bond behavior, high temperature, active prestressing, active fibers

## Abstract

This study investigated the bond behavior and radial crack between concrete and reinforcing bars using cold-drawn shape memory alloy (SMA) crimped fibers controlled by the temperature and volume fraction of the fibers. In this novel approach, the concrete specimens containing cold-drawn SMA crimped fibers with 1.0% and 1.5% volume fractions of cold-drawn SMA fibers were prepared. After that, the specimens were heated to 150 °C to generate recovery stress and activate prestressing within the concrete. The bond strength of specimens was estimated by pullout test using the universal testing machine (UTM). Furthermore, the cracking patterns were investigated using radial strain measured by a circumferential extensometer. The results showed that adding up to 1.5% of SMA fibers improved the bond strength by 47.9% and reduced radial strain by more than 54%. Thus, heating specimens containing SMA fibers showed improved bond behavior compared with non-heated specimens with the same volume fraction.

## 1. Introduction

The performance of bond behavior between steel bars and concrete depends mainly on the following three factors: chemical adhesion, friction, and mechanical interlock [1]. Over the past few decades, a series of studies have discovered the technique for improving the bond behavior of bars inside concrete. One of the most effective approaches was shown to be adding appropriate fiber volume fractions to the concrete matrix [2,3,4,5]. In this regard, Harajli et al. implemented 32 beam specimens with various volume fractions of steel fiber-reinforced concrete (SFRC) to investigate the bond–slip behavior of reinforcing bars embedded in plain and fiber concrete [4]. The results showed that concrete-added fibers with 1 and 2% in volume fractions increased the bond strength, on average, by 26 and 33%, respectively. Furthermore, the steel reinforcing fibers significantly enhanced the ductility of bond failure. Besides the aforementioned steel fiber-reinforced concrete, various types of fibers such as glass fibers (GF) and synthetic fibers were also used for the concrete matrix [6,7,8]. However, all the mentioned fiber types were passive fibers that were only activated due to strain or deformation developed in concrete by external loading. Recently, shape memory alloy (SMA) fibers have been used to reinforce cementitious materials. SMAs are rising as a smart material in civil engineering, and recovery stress due to the shape memory effect of SMAs is significantly useful to enhance the performance of cementitious materials. If the SMA fiber is not activated by heating, the fiber does not induce stress recovery, where the SMA fiber is still a passive one; thus, its performance is similar to that of the steel fiber [9]. However, if the SMA fiber is activated by heating, the active fiber shows outstanding performance in cementitious materials. Several previous studies by Choi et al. [10,11] demonstrated that crimped SMA fibers made by cold drawn NiTi SMA wires showed several outstanding features in cementitious materials. These features include a prestressing effect, crack-closing capacity, enough bond resistance for crack bridging, and availability for mass production. Here, the prestressing effect and crack-closing capacity are due to the recovery stress, and thus, they can be categorized as active actions, while the crack-bridging is a passive action, which works to increase the ductility of cementitious materials. Several types of SMA fibers, namely, paddled, L-, N-, and indent-shaped, have been used [12], while they are not suitable for mass production. The crimped SMA fiber is available for mass production because crimping can be conducted continuously using two gears [13]. A long, crimped wire is firstly prepared, and then it can be cut into short-length portions to make crimped fibers.

For the crimped SMA fiber, its geometries, such as wave depth and length, are critical to bond behavior. Larger wave depth induces better pullout resistance, and the wave height-to-diameter ratio was found to be important for the pullout behavior of crimped SMA fibers [14].

Other recent studies investigating SMA fibers used straight or simple end-deformed fibers, such as straight, end-hooked, and knotted end [15,16]. These SMA fibers are different from the crimped SMA fiber in terms of providing bond resistance. The straight fiber provides bond resistance only due to frictional and chemical bonds, which could be too weak to bear the developed recovery stress in the fiber. The hook-end and knotted-end fibers provide end-anchoring resistance, which is much higher than the friction and chemical bonds. However, the bonds of such fibers do not increase with increasing fiber length. Crimped SMA fiber provides a mechanical bond along the length, which means that the unit length provides a specific bond resistance. Thus, the bond of the crimped fiber can increase with increasing fiber length.

Despite the promising results of SMA fibers, the active effects of SMA fibers has not been investigated in terms of the bond behavior of steel reinforcing bars embedded in concrete. Thus, this study aimed to examine the bond behavior between the deformed steel bar and reinforced concrete by using cold-drawn SMA crimped fibers. Pull-out tests were conducted to obtain experimental results, and a pullout behavior model was developed based on the bond–slip behavior of the pullout test specimens. To do so, 18 cylindrical specimens, including plain concrete and reinforced ones with SMA crimped fibers, were prepared. For considering the volumetric fraction of SMA fibers, the volumetric ratios were controlled by 0%, 1.0%, and 1.5%. Additionally, each type of specimen had two cases of non-heating and heating to investigate the active effect of recovery stress on the pullout behavior of a deformed steel bar.

## 2. Specimens

### 2.1. Crimped SMA Fibers

In this study, Ni50.4-Ti(wt.%) SMA wires were used and then cold drawn to induce pre-strain by themselves. The cold-drawing process was previously described in other studies [13,17]. Briefly, SMA wires were annealed to straighten them and remove disruptions before cooling to room temperature. After that, the cold-drawing procedure started by heating the SMA wires to the transformation temperature and then decreasing the temperature of the wires to room temperature. The final diameter of the SMA wire used in the test was 0.81 mm. In the next step, a rolling device crimped them to produce crimped wires with a wave depth of 0.081 mm. The wave depth was chosen as approximately 10% of the diameter to avoid yielding on the SMA crimped fiber. As a result, nonlinear actions occurred with unpredictable behavior on concrete [13]. After crimping, the diameter of the crimped wires was almost stable [18]. The SMA fibers were produced by cutting the SMA wire to a length of 30 mm. Thus, the ratio of fiber length (l_f_) to fiber diameter (d_f_) was 37.04. The length and l_f_/d_f_ ratio were chosen to be close to those of the previous study [19]. Garcia-Taengua et al. showed that specimens using straight fibers with a length of 35 mm and l_f_/d_f_ ratio of 45 provided better bond behavior than another length and l_f_/d_f_ ratio. The dimensions of SMA crimped fibers are presented in Table 1 and Figure 1. The fibers had the yield stress and ultimate stress of 920 MPa and 1100 MPa, respectively. The secant modulus for the fibers was 15.8 GPa.

The thermo-analytical technique called differential scanning calorimetry (DSC) was used to determine the starting austenite temperature (A_s_) and finishing austenite temperature (A_f_) of the SMA crimped fibers. The results were 48.05 °C and 110.16 °C, respectively (as shown in Figure 2). With the starting temperature of austenite (A_s_) at 48.05 °C, SMA crimped fibers were perfectly retained at room temperature without any transformation.

Figure 3 shows the recovery stress–temperature relationship of the SMA crimped fibers when heating from room temperature to 100 °C, 200 °C, and 300 °C, respectively. After that, the SMA crimped fibers were cooled down to the initial temperature. The results showed that the recovery stress of the SMA crimped fibers did not increase significantly when heated over 200 °C. This result is in accordance with those of previous studies [20]. The target temperature of 150 °C almost completed the phase transformation. Therefore, after heating to 300 °C, the recovery stress was reduced since the thermal expansion exceeded the contraction because of the shape memory effect.

### 2.2. Specimens

The composition of the concrete matrix included Portland cement, a coarse aggregate with a maximum size of 20 mm, silica sand, and water, as seen in Table 2. During mixing, the various volume fractions of SMA crimped fibers were added based on the mixed design of specimens.

This study used specimens of concrete cylinders with dimensions of 100 mm × 200 mm (D × L). A reinforcing bar with a total length of 350 mm was prepared and placed at the center of the specimens. The embedded length of the bar was 200 mm, with 25 mm of length at the top and bottom of the specimens wrapped with oil paper. This oil paper was used to prevent cone failure and stress concentration on the top and bottom surfaces of the specimens. In addition, the protruded part measured 150 mm beyond the top surface of the specimens. The reinforcing bar diameter (d_b_) was 22 mm, and the clearance distances between the ribs were 14.7 mm. The estimated yield strength was 400 MPa. Based on previous studies with equivalent specimens and methods [21] and because the bond length was small, it was expected that the bar would slide inside the concrete matrix during the pullout test before the yielding reinforcing bar. With a 22 mm bar diameter, the concrete cover (c) was 39 mm. Thus, the ratio of concrete cover-to-bar diameter (c/d_b_) was 1.77. Using this c/d_b_ ratio, the study aimed to simulate the bond behavior in a flexural structure more accurately than the well-confined specimens. Moreover, this c/d_b_ ratio helped to avoid the effect of sufficient concrete cover to the failure mode with a c/d_b_ of more than three [22]. Finally, the concrete matrix was cast into the plastic molds and cured for 28 days in water conditions.

In total, 18-cylinder specimens for the pullout test were divided into three groups of SMA crimped fiber volume fractions (V_f_). Each group had six specimens, with half of them heated in an oven at 150 °C for 24 hours to activate the shape memory effect (SME). The series of prepared specimens is shown in Table 3. After turning the drying oven off, the specimens gradually reached room temperature and were prepared for the pullout test. The names of the specimens indicate the volume fraction of the SMA crimped fibers and temperature conditions. The symbols “P” and “CR” indicate plain and SMA crimped fiber concrete. “N” and “H” denote non-heating and heating, respectively. For illustration, “CR10-N-1” expresses that the specimen contained cold-drawn SMA crimped fibers with a 1.0% volume fraction, had a non-heating temperature of 25 °C, and was the first sample. The compressive strength test on the cylindrical specimens was based on the requirements of ASTM C39 [23].

This study used relatively small cylinders compared with those of 150 mm × 300 mm (D × L). A previous study showed that the pullout failure mode of small concrete cylinders was splitting failure with the same steel bar diameter, while large concrete cylinders showed pullout failure of the bond strength. It is conceivable that the SMA fibers’ contribution to bond behavior could be restricted in the pullout failure because the pullout failure depends on the concrete shear capacity. Differently, for the splitting failure, concrete tensile strength is critical to the bond behavior. Activated SMA fibers are proven to enhance tensile strength and ductility. Thus, SMA fibers could be effective in the splitting failure mode of concrete.

## 3. Test Setup and Experiment

Pullout tests were performed on the cooled specimens to evaluate the bond behavior between steel rebar and concrete containing SMA crimped fibers. The specimens were placed on a specialized steel load-carrying framework, with two rigid rectangular plates on the top and bottom with a 40 mm hole in the center of the plate. The protruded reinforcement bar was pulled out by an actuator through the hole due to loading. The linear voltage displacement transducer (LVDT) was set up at the bottom of the bar to measure the slip of the bar. An extensometer was placed around the perimeter in the middle of the specimen to measure the circumferential expansion of the specimen, as shown in Figure 4. The reinforcement bar slip was used to determine the bond stress and bond–slip behavior. The circumferential expansion was used to calculate the splitting strain and investigate the splitting behavior. All tests were conducted by displacement control mode with a test speed of 1.0 mm/min until the specimens failed.

## 4. Results and Discussion

### 4.1. Failure Modes and Cracking

The pullout test results of plain and reinforced specimens are shown in Table 4. The pullout force (F) applied to the specimen can be converted to the average bond strengths (τ_b_) along the effective bond length of the reinforcing bar (L_b_) by using Equation (1). This equation assumed that the bond stress of a reinforcing bar in concrete was distributed uniformly along the bar length.
(1)$$\tau_{b}=\frac{F}{{\pi d}_{b}L_{b}}$$

The radial strain ε_r,p_ at the peak bond strength was calculated using Equation (2), where ∆ is the circumferential expansion measured by circumferential extensometers and D is the diameter of the specimen.
(2)$$\varepsilon_{r,p}=\frac{\Delta }{\pi D}$$

Based on previous studies and codes [4,22,24], the bond strength at failure was proportional to $\sqrt{f{{}^{\prime }}_{c}}$. Hence, Table 4 includes the bond strength normalized to the $\sqrt{f{{}^{\prime }}_{c}}$ presence in the seventh column for investigating bond behavior. The results also include the maximum slip (S_p_). Note that the pullout force was lower than the yield strength of the reinforcing bar in all cases.

The average concrete compressive strength ${\left({f^{\prime }}_{c}\right)}_{ave}$ of the specimens is shown in the sixth column of Table 4. The results showed that increasing the volume fraction of SMA crimped fibers decreased the compressive strength of concrete on non-heating specimens. The average compressive strength of the plain specimens and those containing 1.0% and 1.5% fibers decreased by 13.98% and 17.54%, respectively. For more details, Figure 5 shows the SMA fiber’s effect on the compressive strength by non-heating (N-H) and heating (H) conditions. The negative effects of the SMA crimped fiber on the compressive strength were similar to the concrete containing steel fibers observed in previous studies [25,26]. However, the heated specimens maintained the same compressive strength between plain and SMA crimped fiber-reinforced concrete. This finding is contrary to those of previous studies using steel fibers [2,5]. As a result of heat, steel fibers reduced the compressive strength of the concrete. Hence, the SMA crimped fibers induced recovery stress with an increasing temperature due to the SME. As a result, recovery stress was provided, and prestressing inside the concrete seemed to increase its compressive strength.

Figure 6 shows the cracking patterns of plain and SMA crimped fiber-reinforced specimens. Based on observing the failure pattern and the bond stress–slip graph shape, the specimen’s failure mode was observed to be the splitting mode in all cases.

The splitting failure mode occurred due to the development of radial stresses leading to an extension between the concrete and reinforcing bar ribs. As a consequence, bottom and side longitudinal cracks formed and caused damage to the specimens. The cracking of specimens related to the radial strain was determined by the circumferential extensometer shown in the fifth column in Table 4. All reinforced concrete specimens had visible cracks at the peak value of bond stress. The cracking of CR10-N was closely similar to CR15-N. The side-splitting cracks developed in a partial pattern and did not extend along the full length of the specimens. It was also observed that the cracking patterns of CR10-H were similar to those of CR10-N and CR15-N. However, the cracks developed slower than in non-heating-reinforced specimens. However, the crack expansion of CR15-H specimens developed rapidly, and the crack developed on the full length of the specimens, as shown in Figure 6.

The interior views of the plain and SMA crimped fiber-reinforced specimens are shown in Figure 7. The surface of plain concrete contacting the bar was clear as the bar slipped on the surface. Meanwhile, the SMA crimped fiber specimens showed cracking in the interaction surface between the concrete and the bar ribs. Moreover, an experiment in a previous study compared the interior view with confined specimens [19] by using the external SMA wire jacket method. Figure 7b shows the radial crack that gradually developed and its surrounding distribution. Furthermore, the contacting surface was rough and hard to detach. This behavior was contrary to that of the external SMA wire jacket, which almost crumbled after releasing the jacket with a visible crack. Thus, different behavior was observed due to the crack-bridging effects of the fibers inside the concrete matrix.

### 4.2. Bond Stress–Slip Relationship

The relationship between the average bond stress–slips of specimens affected by the SMA crimped fibers’ volume fractions is shown in Figure 8.

The bond stress–slip graph of plain concrete specimens typically dropped after reaching the peak bond strength. Meanwhile, the SMA crimped fiber-reinforced specimens showed a horizontal branch after descending from the peak bond strength. Moreover, the bond strength was not only dependent on the SMA crimped fibers’ volume fraction, but also reliant on different temperature conditions. For the non-heating specimens shown in Figure 8a, the average peak bond strength values of P-N, CR10-N, and CR15-N were 5.73 MPa, 6.49 MPa, and 8.34 MPa, respectively. In addition, the corresponding average slips were 1.411 mm, 1.155 mm, and 0.923 mm, respectively. Thus, all test series consistently increased in the bond strength with increasing SMA crimped fiber reinforcement, and the slip at the peak point strength was smaller when more SMA crimped fibers were added to the specimens compared with the plain specimens. Furthermore, Figure 8b shows the heating specimens. The average bond strength values of P-H, CR10-H, and CR15-H were 3.97 MPa, 5.47 MPa, and 5.88 MPa, respectively. The corresponding average slips were 1.187 mm, 1.117 mm, and 1.195 mm, respectively. Thus, the effect of SMA crimped fibers on the bond strength of the specimens was positive in all cases. However, the slip behavior of the SMA crimped fibers reinforced by heating decreased from P-H to CR10-H. After that, the slip behavior ramped up from CR10-H to CR15-H by around 7%. Figure 8 also shows that the specimens in the same temperature conditions can be considered to have a similar bond–slip behavior before cracking (i.e., pre-cracking) developed inside the specimens, whether with or without SMA crimped fibers.

The effects of temperature on the bond stress–slips are shown in Figure 9. Correspondingly, temperature’s effect on bond strength is shown in Figure 10. The bond strength was significantly reduced when exposed to high temperatures. The bond strength of plain concrete specimens showed the lowest residual bond strength after heating at 69.4%. Meanwhile, the CR10-H and CR15-H specimens with the addition of SMA crimped fibers showed higher residual bond strengths of 84.3% and 70.5%, respectively. The decrease in bond strength due to heating is because of deterioration in the mechanical properties of concrete upon heating [27,28]. However, the SMA fiber specimens with heating showed higher residual bond strength than plain concrete specimens since the effect of prestressing affected the adhesive in the concrete matrix. As aforementioned, this effect improves compressive strength, i.e., the crucial factor that affects bond strength behavior. In addition, the CR15-H specimen showed lower residual strength than CR10-H because of some abovementioned experimental errors.

In further detail, we used the reference data from previous research to evaluate the residual bond strength of the SMA crimped fiber-reinforced specimens, as shown in Table 5 and Figure 11. Previous studies report that the residual bond strength is mainly affected by the ratio of concrete cover-to-bar diameter (c/d_b_), i.e., larger c/d_b_ ratios will lead to larger residual bond strength [5,29]. Furthermore, the longer annealing time of specimens at high temperatures also causes more destructive chemical behavior of concrete [30]. As shown in Table 5, this study used the smallest c/d_b_ ratio of 1.77 and the longest time, 1440 minutes, for the treatment of specimens at high temperatures. Therefore, this study showed the smallest residual bond strengths compared with previous researchers in terms of the plain concrete specimen. For SMA crimped fiber-reinforced specimens, the bond strength was assessed with steel fiber specimens in previous research because SMA fiber was a novelty and because of a lack of reference data in bond behavior. Generally, besides the effects of the ratio of concrete cover to bar diameter and annealing time, previous research reported that adding steel fiber within 2% of the volume fraction improved the bond behavior in similar temperature conditions (Bengar et al. [5]). However, steel fibers were distributed randomly in the concrete matrix and were difficult to control similarly in all specimens. Therefore, experimental errors occurred due to test implementation (1% SF + NC specimens, Fakoor et al. [2]). In this study, the CR10-H specimen showed the highest residual bond strength at 84.3%, i.e., 14.9% higher compared with the P-H specimen. However, the CR15-H showed a 1.1% higher bond strength than P-H (as shown in Figure 11). The non-significant effects of SMA fibers on the CR15-H specimens seem to be some of the aforementioned experimental errors that were reported in previous studies. Still, specimens containing SMA crimped fibers by heating also had a bar deformation caused by the heating flow through the top and bottom of the specimens.

In addition, the SMA crimped fiber-reinforced specimens by heating showed much more ductile behavior than non-heating SMA crimped fiber specimens, which demonstrated a much more gradual decreasing stage after reaching the peak bond stress.

To investigate the bond–slip behavior further, the pullout prediction behavior model was developed and compared with the previous prediction models, which were utilized based on the model proposed by Ciampi et al. [31] and other researchers [32,33]. Based on the study, the nonlinear pullout behavior model of the bond stress–slip for the deformed bar in concrete is expressed in Equation (3). In addition, the schematic bond behavior of well-confined concrete (W), moderate-confined concrete (M), and unconfined concrete (U) is shown in Figure 12. In this graph, the well-confined concrete (W) corresponds to the pullout failure mode with the shape of failures showing four parts, including an ascending part to τ_1_(W) corresponding slip S_1_, a horizontal at the maximum bond stress τ_1_(W), a descent to τ_3_(W), and a horizontal branch at bond stress τ_3_(W), in order. The moderate-confined (M) and unconfined concrete (U) correspond to the splitting failure mode with a disappeared flat part at the peak bond stress. Unconfined concrete also lacks a horizontal branch after the downward part.
Figure 11Comparison of the previous research and experimental results on residual bond strength [2,5,32].
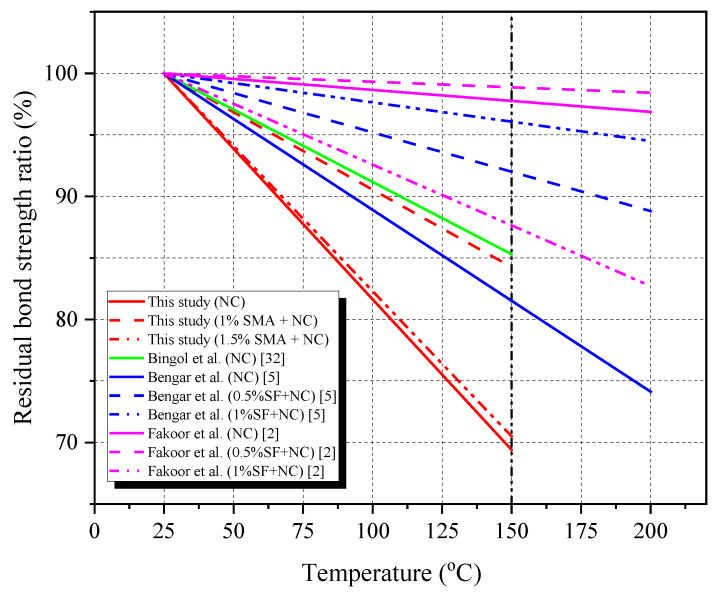

materials-16-01947-t005_Table 5Table 5Previous research of residual bond strength after specimens were heated.Researcherc/d_b_Treatment Duration Bond Strength at Room Temperature Residual Bond Strength Ratio(mins)(MPa)(%)This study (NC)1.7714405.7369.4 (150 °C)This study (1% SMA + NC)1.7714406.4984.3 (150 °C)This study (1.5% SMA + NC)1.7714408.3470.5 (150 °C)Bingol et al. (NC) [32]5.7518019.0685.3 (150 °C)Fakoor et al. (NC) [2]4.56025.5196.9 (200 °C)Fakoor et al. (0.5% SF + NC) [2]4.56024.6798.4 (200 °C)Fakoor et al. (1% SF + NC) [2]4.56021.2282.7 (200 °C)Bengar et al. (NC) [5]2.256012.9174.1 (200 °C)Bengar et al. (0.5% SF + NC) [5]2.256012.4788.8 (200 °C)Bengar et al. (1% SF+NC) [5]2.256012.2094.5 (200 °C)“NC” means normal concrete; “SF” means steel fiber.
(3)$$\tau =\tau_{1}{\left(\frac{s}{s_{1}}\right)}^{\alpha }$$

In Equation (3), valid for 0 ≤ s ≤ s_1_, τ_1_ is presented as the peak of bond strength. For s_1_ ≤ s ≤ s_2_, the bonding stress plateaued as τ = τ_1_ followed by the pullout failure mode when the splitting failure mode was s_1_ = s_2_. After that, τ decreased to the ultimate frictional bond stress τ_3_ at a slip value of s_3_. The values s_1_, s_3_, τ_1_, τ_3_, and α were chosen to match the experimental test.

Table 6 shows some of the previous work models developed for unconfined and confined concretes, which satisfy both of those utilized in Equation (3), and the predicted bond strength was normalized to $\sqrt{f{{}^{\prime }}_{c}}$. The specimens failed in the pullout and splitting modes in terms of the quality of the concrete conditions.

The typical equation for the prediction bond strength τ_1_, τ_3_ shown in Table 6 was mentioned by some previous researchers [35,36] given as:(4)$$\tau =\beta \sqrt{f{{}^{\prime }}_{c}}$$

In Equation (4), the coefficient β was formulated depending on various factors such as temperature conditions, the ratio of the concrete cover to the rebar diameter, or fiber volume fractions. Based on the predicted bond strengths τ_1_ and τ_3_, as shown in Table 6, the β value for the pullout failure mode varied in the range of 1.25–2.57 and 0.5–1.0, respectively.

By corresponding to the experiment bond stress-slip data of this research and the model proposed in Equation (3), the proposed behavior parameters are given in Table 7. Note that the bond strengths τ_1_ and τ_3_ for pullout specimens were predicted by using Equation (4) for the corresponding value β that is close to the value shown in the seventh column of Table 4. Moreover, the bond behavior model is shown in Figure 9. It was observed that the predicted behavior model properly captures the bond–slip behavior of the pullout specimens. As a result of predicting the bond strength τ_1_, the values of β for P-N and P-H were 0.9 and 0.6, respectively. For the SMA crimped fiber-reinforced concrete specimens, the values of β for CR10-N, CR15-N, CR10-H, and CR15-H were 1.1, 1.5, 0.9, and 0.9, respectively. Another point worth mentioning, as shown in Table 7, is the curve fitting parameter α. This coefficient changes from 0 to 1 depending on the bond characteristics. Additionally, the predictions of the bond stress graph before reaching the peak strength were more linear with a larger value of α [22,37]. The prediction value α was 0.8 for plain and passive fiber specimens, and 0.6 for active fiber specimens. Thus, the active fibers showed more ductile behavior than plain and passive fibers in reinforced concrete specimens. However, the α value was larger than some previous studies’ results, as shown in Table 6, with values ranging from 0.3 to 0.4. Moreover, compared with the bond strengths predicted in previous studies, as shown in Table 6, the result was lower in all cases. Hence, the specimens containing SMA crimped fibers reinforced showed the moderate-confined concrete more than well-confined concrete. Note that the well-confined concrete shown in Table 6 was applied to concrete covers larger than 5 dp or external jackets.

A comparison of the average bond strength ratio between specimens containing 1.0% and 1.5% volume fraction SMA crimped fibers to plain concrete in terms of the temperature is shown in Figure 13. For non-heating specimens, the bond strength ratio changed from P-N to CR10-N by 13.3%. After that, increasing the ratio of SMA crimped fibers from 1.0% to 1.5% massively increased the bond strength by 28.5%. Thus, the SMA crimped fibers also provided passive confinement to the concrete. This phenomenon was also observed in previous studies that used steel fibers, which increased the fibers’ volume fractions by 1% and 2% and increased the bond stress by 26% and 33%, respectively [4]. The bond strength ratio increased rapidly for heating specimens by adding SMA crimped fibers with 0% to 1% of volume fractions by 37.6%. This result was higher by 24.2% for the CR10-N specimens. Thus, the heating provided an active prestressing effect inside the concrete matrix. This behavior caused by a high temperature in SMA fibers induced recovery stress due to the SME. However, the bond strength ratio from CR10-H to CR15-H slightly increased by 7.5%. The effect of active prestressing in this case was not shown clearly.

### 4.3. Relationship between Bond Stress and Radial Strain

The relationships between the bond stress and the circumferential strain in terms of temperatures are shown in Figure 14. For the plain specimens, the radial strain increased rapidly after the formation of cracks and developed around the peak bond stress. The radial strain of the P-N and P-H at the bond strength was 2.94 × 10^−4^ and 7.1 × 10^−4^, respectively. Thus, the plain concrete specimens that were heated showed more than two-fold negative effects of radial strain compared with the P-N specimens.

For non-heating SMA crimped fiber-reinforced specimens, the radial strains were stable with the average radial strain at the peak bond stress of CR10-N and CR15-N being 4.5 × 10^−4^ and 4.81 × 10^−4^, respectively. However, the heating-reinforced specimens showed a different trend in terms of the radial strain. The CR10-H showed a 54% better performance compared with P-H specimens. In contrast, the CR15-H showed the most considerable radial strain with a value up to 24.25 × 10^−4^. This value was 5.2 times greater than that of CR10-H. As previously stated, the negative effects of SMA crimped fibers on CR15-H specimens were due to some experimental errors. In addition, an abundance of active prestressing caused micro-cracking, which should be investigated in future studies. The bond stress–radial strain response in the post-cracking stage gradually decreased for all reinforced specimens. However, the downward trend of heating-reinforced specimens lingered more than the non-heating-reinforced specimens.

Furthermore, Figure 15 shows the circumferential strain as a function of the slip effect of the SMA crimped fibers’ volume fractions. The radial strain of all specimens generally developed rapidly after slipping over the slip at bond strengths. The plain concrete showed the most extensive slopes compared with SMA crimped fiber-reinforced concretes in all cases. As shown in Figure 14d,f and Figure 15, the heating-reinforced specimens’ radial strains developed a smoother trend than non-heating-reinforced specimens before reaching the peak bond strength.

## 5. Conclusions

This study obtained experimental results from pullout tests of concrete containing cold-drawn SMA crimped fibers. These approaches considered the effects of elevated temperature, changes in the mix design, and differences in cracking patterns. As a result, the bond stress and slip relationships for plain and SMA crimped fiber specimens were compared with the results of the pullout behavior model. Furthermore, this study also observed the radial strain using a circumferential extensometer in the middle of the specimens. Based on the experimental results, the following conclusions were drawn:The non-heating SMA crimped fiber specimens showed passive bond stress–slip behavior which matched that of the steel fibers in previous studies. The CR10-H showed a 24.2% higher bond strength ratio compared with CR10-N. The CR15-H specimen did not show clear prestressing effects during tests;The bond strength increased with an increased SMA fiber volume fraction under the same temperature conditions. The P-N specimens showed higher bond strength development compared with the P-H specimens. The specimens that contained SMA crimped fibers displayed moderate confinement behavior, unlike the plain specimens, which showed unconfined concrete behavior. The heated SMA fiber-reinforced concrete also showed more ductile behavior in bond strength–slip behavior than non-heated specimens;The radial strains at the peak bond stress showed differing trends in heated and non-heated specimens. The non-heating SMA crimped fiber specimens showed a similar radial strain with an increased volume fraction and were larger than plain concrete specimens. The CR10-H decreased radial strains by more than 54% compared with P-H specimens. However, CR15-H increased radial strains, possibly due to experimental errors.

The bond strength prediction can be formulated for concrete containing SMA crimped fibers with the function of the volume fraction, different temperatures, and recovery stress. The consideration of volume fractions from 0% to 1% of the SMA crimped fibers, the variety of the ratio of concrete cover to bar diameter, as well as the elimination of the effects of high temperature on the deformation of the bar at the top and bottom specimens by using a specialized isolated chamber should be investigated in future studies.

## Figures and Tables

**Figure 1 materials-16-01947-f001:**
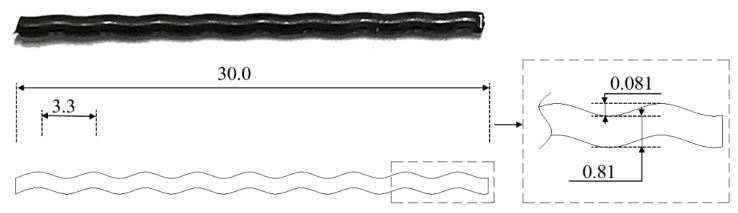
Photography and dimensions of SMA crimped fibers (unit: mm).

**Figure 2 materials-16-01947-f002:**
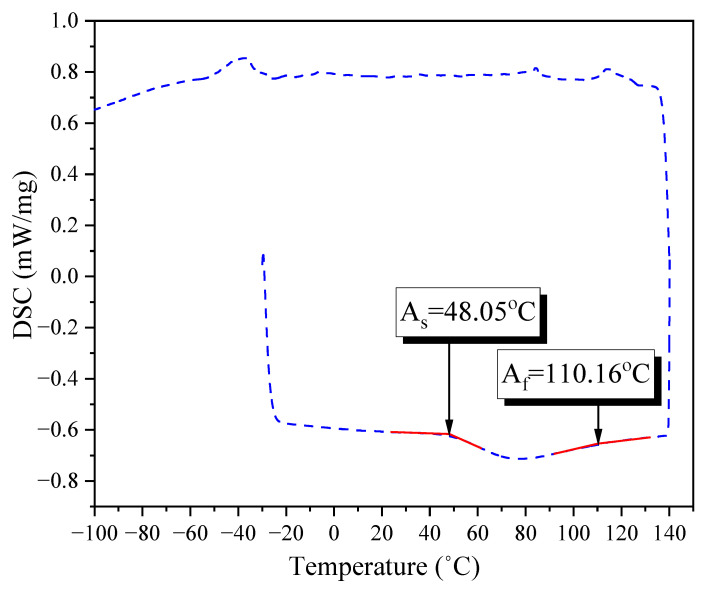
DSC curves of cold−drawn SMA fibers.

**Figure 3 materials-16-01947-f003:**
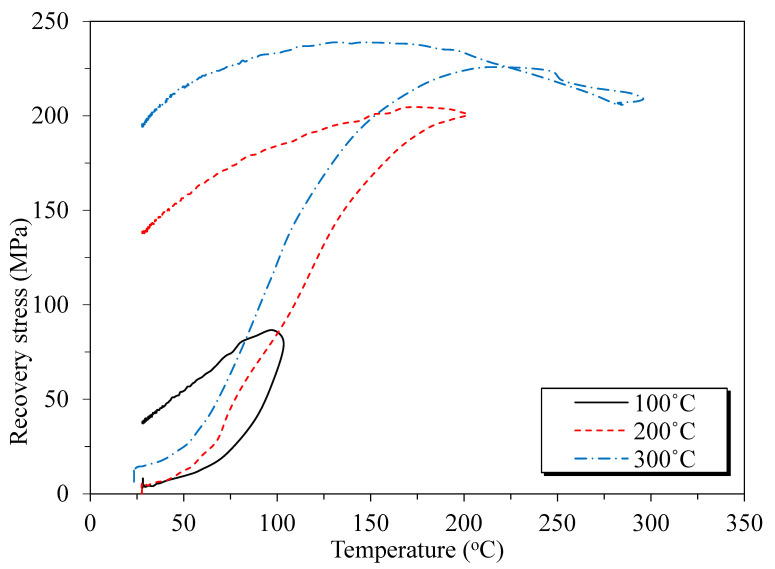
Temperature–stress curves of the SMA crimped fibers.

**Figure 4 materials-16-01947-f004:**
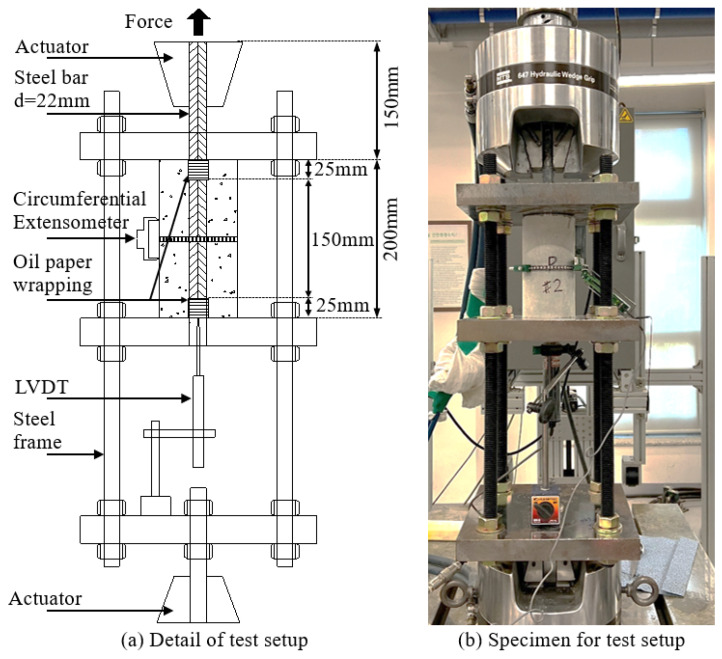
Details of the pullout test setup.

**Figure 5 materials-16-01947-f005:**
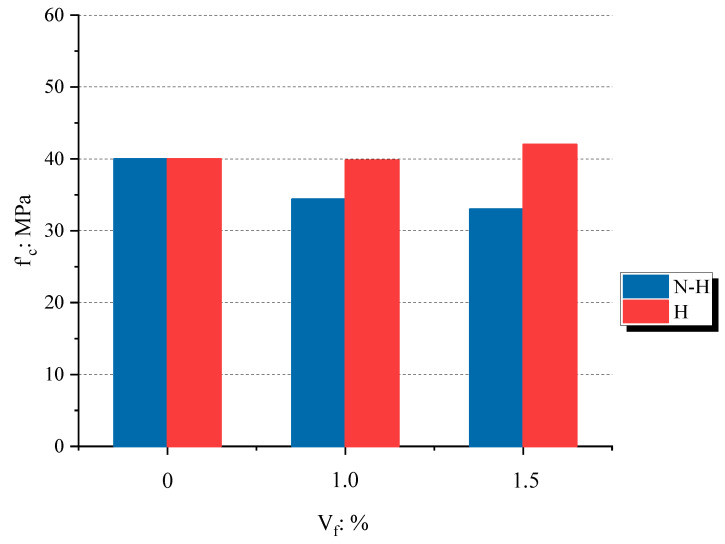
The effect of SMA fiber volume fraction and temperature on the compressive strength.

**Figure 6 materials-16-01947-f006:**
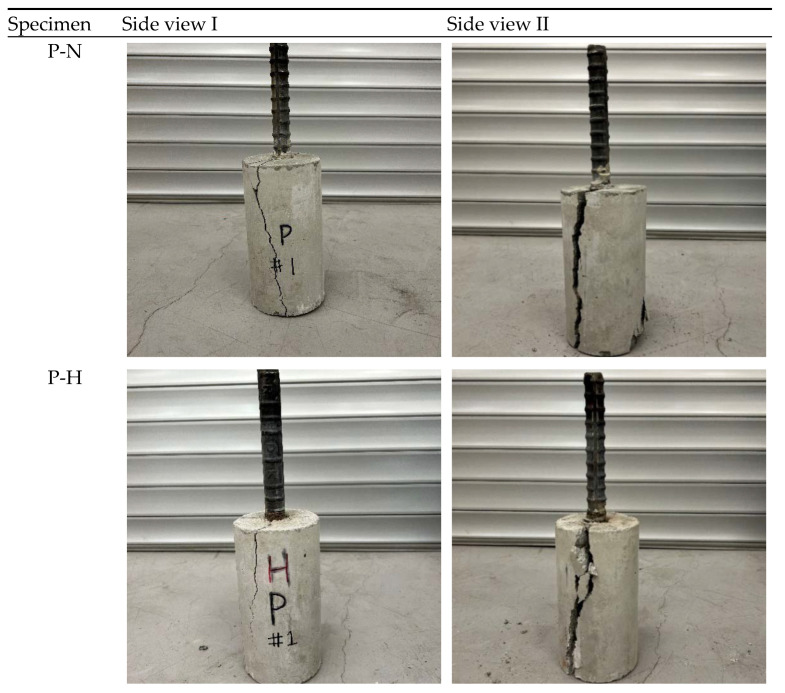
Photo of the failure mode and cracking of the specimens.

**Figure 7 materials-16-01947-f007:**
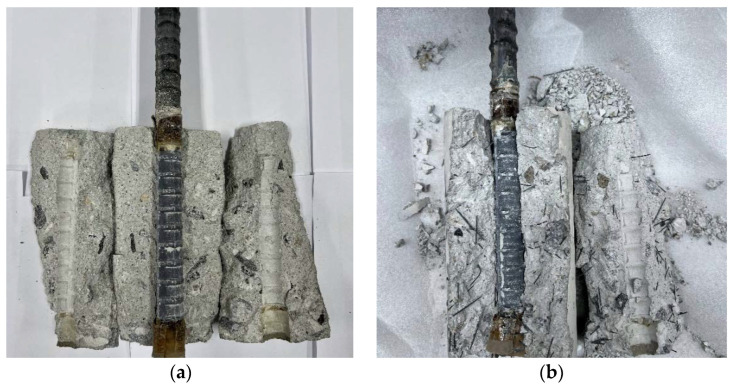
Photo of the inside view (**a**) of plain concrete failure and (**b**) SMA crimped fiber failure.

**Figure 8 materials-16-01947-f008:**
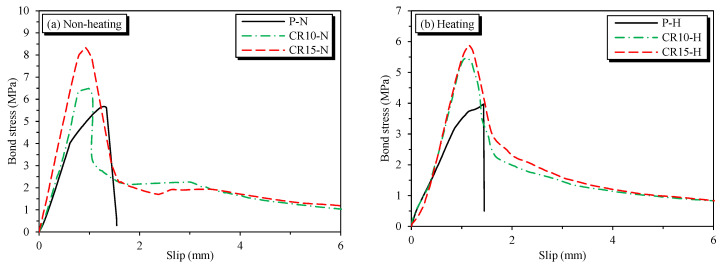
Effect of volume fractions on bond stress as a slip function.

**Figure 9 materials-16-01947-f009:**
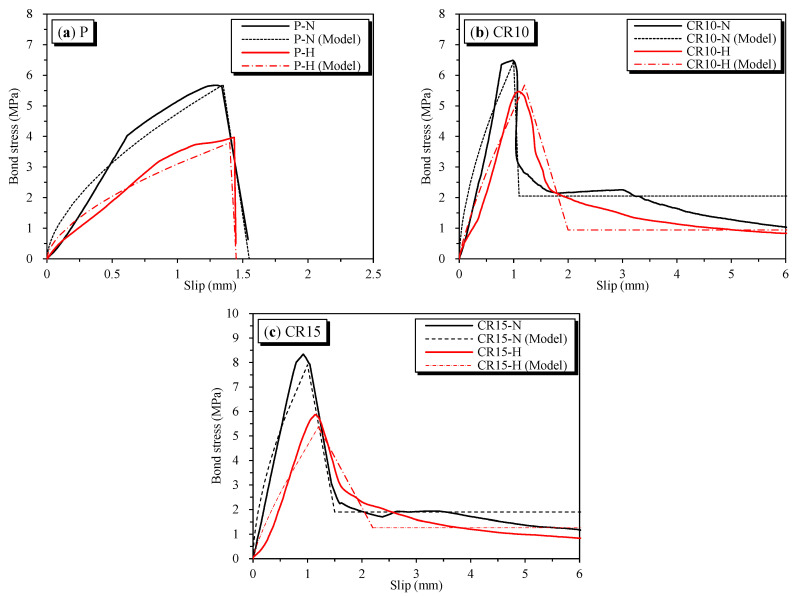
The effect of temperature on the bond stress as a slip function.

**Figure 10 materials-16-01947-f010:**
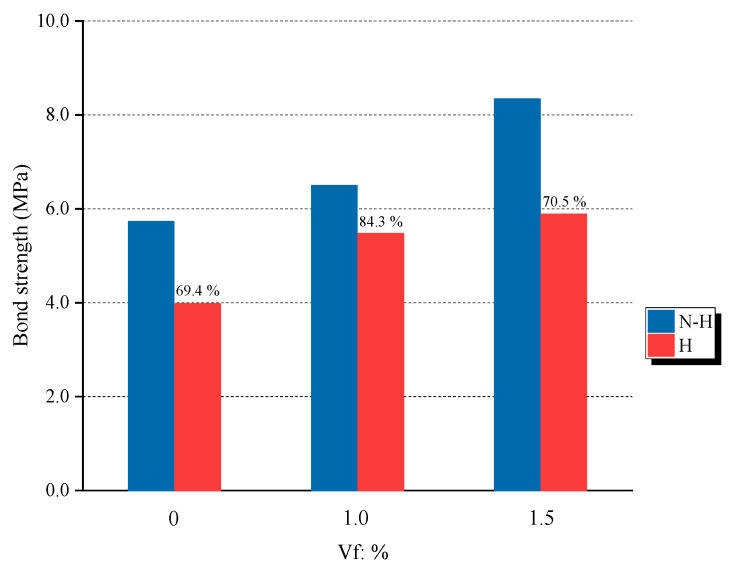
The effect of temperature on the bond strength as a volume fraction.

**Figure 12 materials-16-01947-f012:**
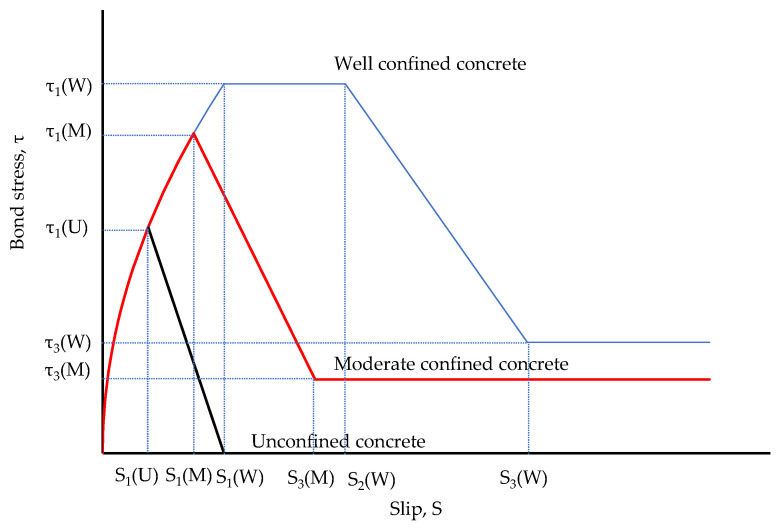
Schematic bond behavior of reinforced concrete.

**Figure 13 materials-16-01947-f013:**
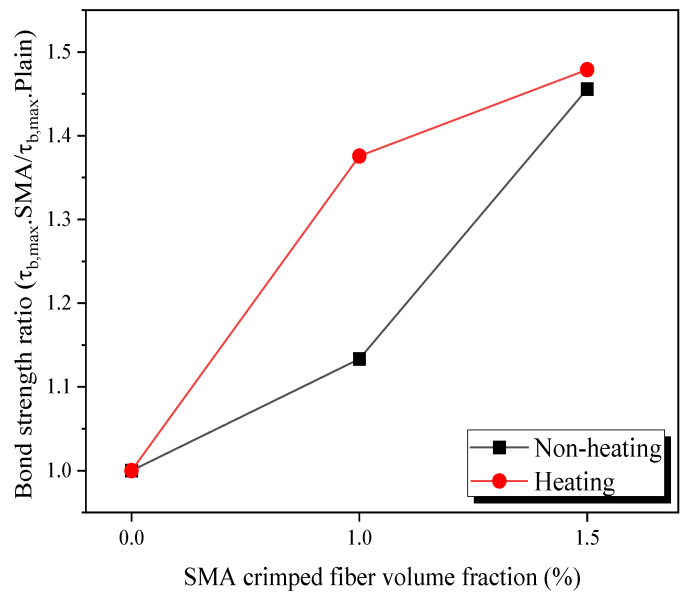
Bond strength ratio between the peak-SMA crimped fibers’ volume fractions.

**Figure 14 materials-16-01947-f014:**
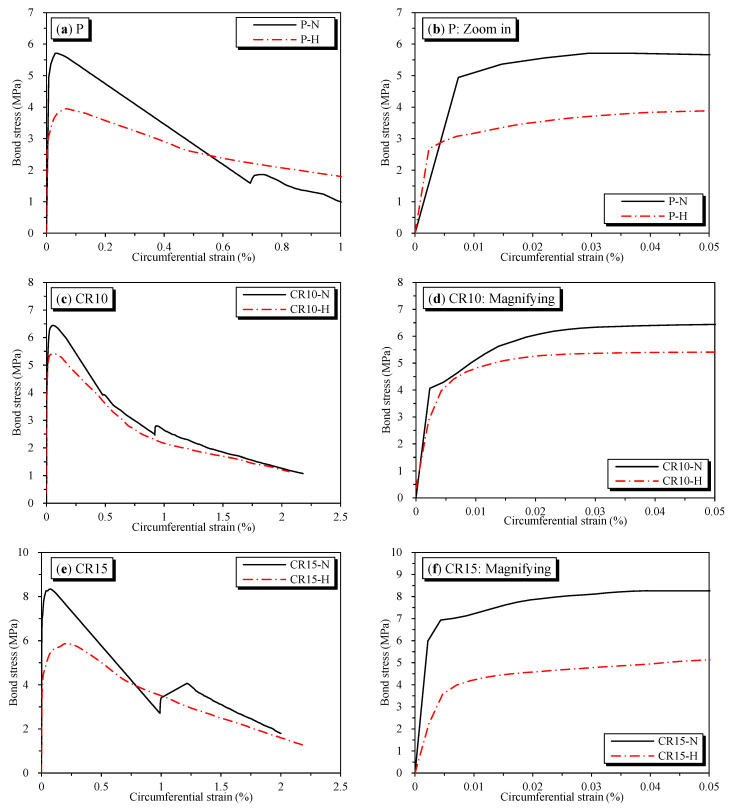
Effect of temperature on the circumferential strain as a function of bond stress.

**Figure 15 materials-16-01947-f015:**
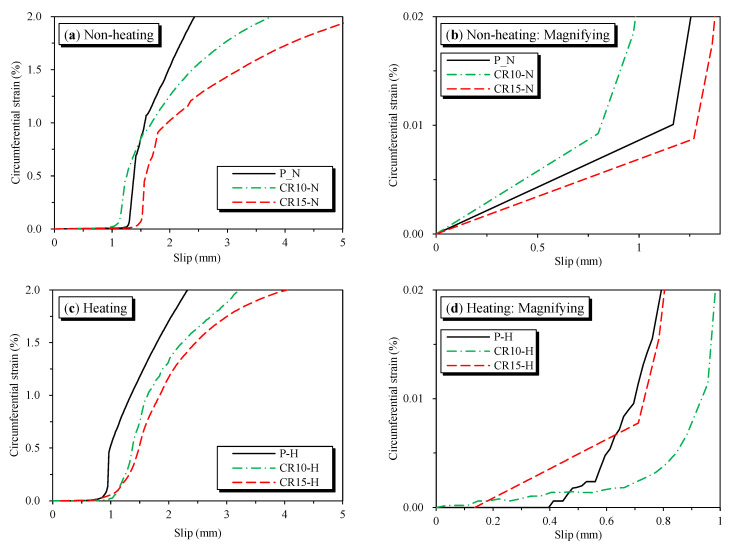
The effect of volume fraction on circumferential strain as a slip function.

**Table 1 materials-16-01947-t001:** Properties of cold-drawn SMA fibers with crimped geometry.

Diameter	Wave Depth	Wave Length	Fiber Length	A_s_	A_f_
(mm)	(mm)	(mm)	(mm)	°C	°C
0.81	0.081	3.3	30	48.05	110.16

**Table 2 materials-16-01947-t002:** Composition of concrete matrix by weight ratio.

Portland Cement	Coarse Aggregate	Silica Sand	Water
1.3	0.6	0.6	0.5

**Table 3 materials-16-01947-t003:** Specimen details.

Specimens	V_f_	Temperature Exposure
	(%)	(°C)
P-N-1	0	25
P-N-2	0	25
P-N-3	0	25
P-H-1	0	150
P-H-2	0	150
P-H-3	0	150
CR10-N-1	1	25
CR10-N-2	1	25
CR10-N-3	1	25
CR10-H-1	1	150
CR10-H-2	1	150
CR10-H-3	1	150
CR15-N-1	1.5	25
CR15-N-2	1.5	25
CR15-N-3	1.5	25
CR15-H-1	1.5	150
CR15-H-2	1.5	150
CR15-H-3	1.5	150

**Table 4 materials-16-01947-t004:** Results from the pullout tests of specimens.

Specimens	F	$\tau_{b,max}$	S_p_	ε_r,p_(×10^−4^)	${\left({f^{\prime }}_{c}\right)}_{ave}\;$	$\tau_{b,max}/{\left({f^{\prime }}_{c}\right)}^{0.5}\;$
	(kN)	(MPa)	(mm)		(MPa)	
P-N-1	61.9	5.97	1.315	3.22	40.0	0.974
P-N-2	57.9	5.59	1.423	2.75		0.884
P-N-3	58.3	5.62	1.494	2.86		0.861
* Average *	* 59.4 *	* 5.73 *	* 1.411 *	* 2.94 *	* - *	* 0.906 *
P-H-1	44.5	4.29	1.155	7.04	40.0	0.648
P-H-2	38.1	3.67	1.166	7.46		0.588
P-H-3	41.1	3.96	1.239	6.80		0.650
* Average *	* 41.2 *	* 3.97 *	* 1.187 *	* 7.10 *	* - *	* 0.629 *
CR10-N-1	70.8	6.83	1.133	4.59	34.4	1.204
CR10-N-2	61.0	5.89	1.265	4.23		0.998
CR10-N-3	70.0	6.75	1.067	4.67		1.120
* Average *	* 67.3 *	* 6.49 *	* 1.155 *	* 4.50 *	* - *	* 1.107 *
CR10-H-1	54.2	5.23	1.133	4.31	39.8	0.855
CR10-H-2	55.6	5.36	1.067	4.87		0.851
CR10-H-3	60.2	5.81	1.150	4.67		0.895
* Average *	* 56.7 *	* 5.47 *	* 1.117 *	* 4.62 *	* - *	* 0.867 *
CR15-N-1	85.1	8.21	0.991	4.73	33.0	1.468
CR15-N-2	83.6	8.06	0.933	4.59		1.457
CR15-N-3	90.6	8.74	0.845	5.12		1.433
Average	86.4	8.34	0.923	4.81	-	1.453
CR15-H-1	66.5	6.42	1.210	24.49	42.0	0.988
CR15-H-2	59.9	5.78	1.065	21.72		0.884
CR15-H-3	56.4	5.44	1.311	26.54		0.848
* Average *	* 60.9 *	* 5.88 *	* 1.195 *	* 24.25 *	* - *	* 0.907 *

**Table 6 materials-16-01947-t006:** Prediction parameters for bond stress–slip proposed by codes and other researchers.

Researcher/Code	Failure	Concrete Condition	s_1_ (mm)	s_2_ (mm)	s_3_ (mm)	α	τ_1_	τ_3_
FIB Model Code [22]	Pullout	Good condition	1.0	2.0	c_clear_	0.4	$2.5\sqrt{f{{}^{\prime }}_{c}}$	$\sqrt{f{{}^{\prime }}_{c}}$
		All other bond conditions	1.8	3.6	c_clear_	0.4	$1.25\sqrt{f{{}^{\prime }}_{c}}$	$0.5\sqrt{f{{}^{\prime }}_{c}}$
Choi et al. [21,34]	Pullout	Steel jacket, NiTiNb SMA wires	1.0	3.0	c_clear_	0.4	$2.5\sqrt{f{{}^{\prime }}_{c}}$	$\sqrt{f{{}^{\prime }}_{c}}$
	Splitting	Plain concrete	0.6	s_1_	1	0.4	$2.0\sqrt{f{{}^{\prime }}_{c}}$	$0.3\sqrt{f{{}^{\prime }}_{c}}$
Harajli et al. [4]	Pullout	Steel fibers	0.15 c_clear_	0.35 c_clear_	c_clear_	0.3	$2.57\sqrt{f{{}^{\prime }}_{c}}$	$0.9\sqrt{f{{}^{\prime }}_{c}}$

c_clear_ is the clear rib spacing.

**Table 7 materials-16-01947-t007:** Proposed parameters for predicting the bond–slip behavior model.

Specimens	Failure Mode	s_1_ = s_2_ (mm)	s_3_ (mm)	α	τ_1_	τ_3_
P-N	Splitting	1.35	1.55	0.8	$0.9\;\sqrt{f{{}^{\prime }}_{c}}$	0
P-H	Splitting	1.4	1.45	0.8	$0.6\;\sqrt{f{{}^{\prime }}_{c}}$	0
CR10-N	Splitting	1.0	1.0	0.8	$1.1\sqrt{f{{}^{\prime }}_{c}}$	$0.35\;\sqrt{f{{}^{\prime }}_{c}}$
CR10-H	Splitting	1.2	2.0	0.6	$0.9\;\sqrt{f{{}^{\prime }}_{c}}$	$0.15\;\sqrt{f{{}^{\prime }}_{c}}$
CR15-N	Splitting	1.0	1.6	0.8	$1.5\;\sqrt{f{{}^{\prime }}_{c}}$	$0.3\;\sqrt{f{{}^{\prime }}_{c}}$
CR15-H	Splitting	1.2	2.2	0.6	$0.9\;\sqrt{f{{}^{\prime }}_{c}}$	$0.2\sqrt{f{{}^{\prime }}_{c}}$

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
