# Peer review of "Active SMA Fibers’ Effect on the Pullout Behavior of a Steel Bar Embedded in Concrete"

_materials, 2023, doi:10.3390/ma16051947_

Round 1

Reviewer 1 Report

The authors investigate the bond behavior between concrete with active SMA fibers and reinforcing bar. The results showed the fibers' effectiveness in enhancing the bond strength. The paper is well written and the results are presented clearly. I suggest accept this paper. Some comments are provided as follows:

-  The mechanisms of the active fibers's effects on the bond behavior need be introduced.

- The results will be more useful if a quantitative model for the bond strength of concrete with SMA fibers is proposed.

- The conclusions can be more concise and better be numbered for each point.

Author Response

The paper is revised with following the comments.

The authors investigate the bond behavior between concrete with active SMA fibers and reinforcing bar. The results showed the fibers' effectiveness in enhancing the bond strength. The paper is well written and the results are presented clearly. I suggest accept this paper. Some comments are provided as follows:

-  The mechanisms of the active fibers's effects on the bond behavior need be introduced.

→ Adding mechanical property as the line:  95-97.

- The results will be more useful if a quantitative model for the bond strength of concrete with SMA fibers is proposed.

→Keep opinion. This study formulated a bond behavior model. However, the quantitative model needs to consider more volume fraction value (next study) to consider exact affect behavior.

- The conclusions can be more concise and better be numbered for each point.

→ Updated as the line: 456-472

Thanks.

Choi

Reviewer 2 Report

REVIEW

on article

Active SMA fibers’ effect on the pullout behavior of a steel bar embedded in mortar

Eunsoo Choi, Bui Thanh Nhan and Jonhkeon Choi

SUMMARY

The article submitted for review is relevant and of interest to readers in the field of building materials science. The article examines the behavior of fibers-reinforced concrete using crimped cold-drawn shape memory alloy (SMA). It is expected that the SMA fibers will develop a restorative stress when heated, which activates the prestressing effect within the concrete. This approach is new and of interest to researchers.

The shape memory effect in metals, the discovery of which is rightfully regarded as one of the most significant achievements in materials science in recent years, is currently being intensively studied and, in a number of cases, successfully applied in engineering, construction, medicine, and other industries. The use of the shape memory effect allows solving many technical problems. For example, the creation of hermetic joints, the design of "super springs" and accumulators of mechanical energy, stepper motors, the creation of joints from dissimilar materials (metal-non-metal), when the use of welding or soldering becomes impossible.

In their study, the authors added cold-drawn crimped SMA fibers with a volume fraction of 1.0% and 1.5% to mortar and exposed the samples to high temperatures. The specimens were then subjected to a rebar pullout test. According to the results of the experiment, the authors showed that the addition of up to 1.5% of the volume fraction of crimped SMA fibers in the concrete matrix increased the bond strength to 47.9%. In addition, heating samples containing crimped SMA fibers improved adhesion characteristics compared to samples without heating with the same volume fraction.

COMMENTS

1.    The first observation is that the authors did not formulate the scientific problem in the Abstract and Introduction. I mean the scientific deficit in materials science that exists at the present time. Authors should rearrange the beginning of the Abstract. Editors strongly recommended authors should follow the style of structured abstracts, but without headings: 1) Background: Place the question addressed in a broad context and highlight the purpose of the study; 2) Methods: Describe briefly the main methods or treatments applied. Include any relevant preregistration numbers, and species and strains of any animals used. 3) Results: Summarize the article's main findings; and 4) Conclusion: Indicate the main conclusions or interpretations. Then the abstract will fully meet the requirements of the Materials journal.

2.    The authors presented a literature review in the Introduction. At the same time, it looks shallow, because there is not enough quantitative analysis obtained by other researchers for such a topic problem as SMA. Then there will be a clear understanding of the scientific novelty of the study.

3.    The References list does not match the requirements of the Materials journal. In addition, please provide with DOI every source.

4.    What were the considerations for choosing the dimensions of the crimped SMA steel fiber?

5.    Figure 2 is earlier than the mention of it. Please, disclose the term DSC before the Figure 2 appears.

6.    I recommend the authors add more information for the selected materials in section 2.

7.    In the title of the article, the authors mention "mortar". The text of the article talks about "concrete". It's confusing. Please clarify, since the presence of coarse aggregate changes the local stress-strain state and may change the behavior of the material.

8.    The methodology is described by the authors in sufficient detail and interesting, but I would recommend authors add more of the analytical part to make the article more interesting and perceived by the reader (if it is possible).

9.    In addition, smoother transitions between sections are required. For example, between sections 2 and 3, a smoother transition is needed.

10. The results presented by the authors are interesting but need further explanation. In particular, more detailed explanations are needed for the graphs in Figures 9, 10 and 12.

11. The authors presented a detailed comparison of the obtained results with data from other studies. I recommend that the authors move this part into a separate Discussion section.

12.  In general, the article is interesting, devoted to a topical scientific problem, but needs to be improved. I recommend the article for publishing after corrections.

Author Response

The paper is revised with following the comments.

  1. The first observation is that the authors did not formulate the scientific problem in the Abstract and Introduction. I mean the scientific deficit in materials science that exists at the present time. Authors should rearrange the beginning of the Abstract. Editors strongly recommended authors should follow the style of structured abstracts, but without headings: 1) Background: Place the question addressed in a broad context and highlight the purpose of the study; 2) Methods: Describe briefly the main methods or treatments applied. Include any relevant preregistration numbers, and species and strains of any animals used. 3) Results: Summarize the article's main findings; and 4) Conclusion: Indicate the main conclusions or interpretations. Then the abstract will fully meet the requirements of the Materials journal.

→ Rewrote as line: 7-17

  1. The authors presented a literature review in the Introduction. At the same time, it looks shallow, because there is not enough quantitative analysis obtained by other researchers for such a topic problem as SMA. Then there will be a clear understanding of the scientific novelty of the study.

→ Keep opinion. The scientific novelty in this study is investigating the bond behavior of steel reinforcing bars embedded in SMA fiber reinforced concrete with elevated temperature to induce prestressing effect, as mentioned in lines 66-69. (Almost the present paper investigates SMA fiber on cement mortar)

  1. The References list does not match the requirements of the Materials journal. In addition, please provide with DOI every source.

→ Updated

  1. What were the considerations for choosing the dimensions of the crimped SMA steel fiber?

→ Mentioned as the line:  92-94.

  1. Figure 2 is earlier than the mention of it. Please, disclose the term DSC before the Figure 2 appears.

→ Updated

  1. I recommend the authors add more information for the selected materials in section 2.

→ Adding mechanical property as the line:  95-97.

  1. In the title of the article, the authors mention "mortar". The text of the article talks about "concrete". It's confusing. Please clarify, since the presence of coarse aggregate changes the local stress-strain state and may change the behavior of the material.

→ Updated: Concrete.

  1. The methodology is described by the authors in sufficient detail and interesting, but I would recommend authors add more of the analytical part to make the article more interesting and perceived by the reader (if it is possible).

→ Keep opinion. The analytical part will consider the next study with more fiber volume fraction.

  1. In addition, smoother transitions between sections are required. For example, between sections 2 and 3, a smoother transition is needed.

→ Updated as the line:  171-172

  1. The results presented by the authors are interesting but need further explanation. In particular, more detailed explanations are needed for the graphs in Figures 9, 10 and 12.

→ Updated as the line:  287-297; 344-350

  1. The authors presented a detailed comparison of the obtained results with data from other studies. I recommend that the authors move this part into a separate Discussion section.

 → Keep opinion. The current layout also keeps the reader in the flow of contents.

  1. In general, the article is interesting, devoted to a topical scientific problem, but needs to be improved. I recommend the article for publishing after corrections.

Thanks.

Choi

Reviewer 3 Report

Please have a look on the attached file, There are many writing errors in the manuscript. I can only read it upto line 133. Please follow the comments, improve the quality and re-submit the work.

Author Response

The paper is revised with following the comments.

Thanks.

Choi.

Round 2

Reviewer 1 Report

All comments are replied.

Reviewer 2 Report

All my comments were considered, and appropriate corrections were done. The article looks much better. 

I recommend the article for publishing.

Reviewer 3 Report

All set, no issues